# A cross-sectional study into the prevalence and conformational risk factors of BOAS across fourteen brachycephalic dog breeds

**Francesca Tomlinson**[1]*, **Nai-Chieh Liu**[2], **David R. Sargan**[1], **Jane F. Ladlow**[1,3]

1 Department of Veterinary Medicine, University of Cambridge, Cambridge, United Kingdom, 2 School of Veterinary Medicine, Institute of Veterinary Clinical Science, National Taiwan University, Taipei City, Taiwan, 3 Granta Veterinary Specialists Referrals, Linton, United Kingdom

* ft270@cam.ac.uk

## Abstract

Brachycephalic Obstructive Airway Syndrome (BOAS) is known to occur as a common condition in short-skulled (brachycephalic) dogs, but has been intensively studied only in three breeds: the Bulldog, French Bulldog and Pug. This study investigates the frequency and severity of BOAS in a further 14 breeds in the UK pet population: Affenpinscher, Boston Terrier, Boxer, Cavalier King Charles Spaniel, Chihuahua, Dogue de Bordeaux, Griffon Bruxellois, Japanese Chin, King Charles Spaniel, Maltese, Pekingese, Pomeranian, Shih Tzu and Staffordshire Bull Terrier. The respiratory functional grading (RFG) assessment was adapted for use in these breeds, noting respiratory characteristics for 898 dogs in this study. Conformational parameters were measured to analyse the association with BOAS risk. Statistical analysis was performed both comparatively across the 14 breeds and within each breed. Almost every breed in this study had some detectable level of breathing abnormality. Only the Maltese and Pomeranian had no dogs with clinically significant disease. The Pekingese and Japanese Chin, had the highest rates of BOAS with only 10.9% and 17.4% being Grade 0 respectively. Across the whole study population, three factors were significantly correlated with BOAS: higher body condition score, nostril stenosis, and lower craniofacial ratio (more extreme facial hypoplasia). These parameters accounted for 20% of the variation in BOAS status when modelled in multiple logistic regression. It was noted that some extremely flat-faced breeds, for example the King Charles Spaniel, had lower rates of BOAS than expected based on their conformation. Overall, the frequency of BOAS varies considerably by breed. Broadly speaking, more extreme brachycephaly, nostril stenosis and high body condition score are associated with increased BOAS risk. However, with variation of phenotype between the breeds, the findings of this study advocate for a breed-specific approach when tackling the reduction of the disease on a population level.

**Data availability statement:** All relevant data are within the manuscript and its Supporting Information files.

**Funding:** This work was supported by a grant (PNAG/710) from The Kennel Club Charitable Trust (https://www.kennelclubcharitabletrust.org).The funders had no role in study design, data collection and analysis, decision to publish, or preparation of the manuscript.

**Competing interests:** The authors have declared that no competing interests exist.

## Introduction

Brachycephalic obstructive airway syndrome (BOAS) is a chronic disease associated with the brachycephalic (short-skulled) phenotype in dogs [1]. Obstructive anatomical lesions within the upper respiratory tract result in airway narrowing, leading to wide-ranging clinical signs that impact the dog's ability to function normally [2]. The most recognisable sign of BOAS is noisy breathing, but it can also impact a dog's ability to exercise, sleep and cope with heat or stress. The syndrome varies in severity, with more affected dogs having a reduced quality of life [3]. In severe cases, acute exacerbation could result in cyanosis, collapse and even death. BOAS has primarily been reported in the three most popular brachycephalic dog breeds in the UK: the French Bulldog, the Pug and Bulldog [4–6].

The pathophysiology of BOAS is currently described in terms of the structural abnormalities of the upper respiratory tract resulting in airway obstruction and breathing difficulty [2]. There is variation in the presentation of affected dogs, in terms of both clinical signs and obstructive lesion sites [6,7]. Abnormalities in the nasal cavity such as stenotic nares and aberrant turbinates result in impairment of nasal airflow [8–11]. Dogs with nasal obstruction will often switch to mouth-breathing to circumvent this and, when nasal breathing, stertor may be exacerbated by a large and thickened soft palate contributing to the reduction of the nasopharyngeal lumen [12]. Increased respiratory effort to compensate for these obstructions results in increased negative pressure within the airways leading to secondary abnormalities such as tonsillar or laryngeal saccule eversion. Additional factors can contribute to the severity of breathing dysfunction, such as laryngeal collapse or tracheal hypoplasia which primarily occur in Pugs and Bulldogs respectively [6,13]. Stridor, a high-pitched sound, is associated with laryngeal collapse. In severe cases dogs may require treatment from under 12 months old [14]. Surgery is often the treatment of choice for many clinically affected dogs and weight loss, medication and conservative management strategies may be implemented to reduce the impact of ongoing clinical signs [14,15].

Objective assessment of BOAS cases has previously proven difficult due to normalisation of the clinical signs as a feature of the dogs' breed [16]. A respiratory function grading assessment has been developed as a clinical diagnostic tool enabling consistent assessment of patients [17]. The grading system has been validated through the use of whole-body barometric plethysmography (WBBP) indicating objective differences in the respiratory parameters between affected and unaffected French Bulldogs, Pugs and Bulldogs [18]. Clinically affected dogs demonstrate noisy breathing, described in terms of stertor and stridor, and dogs are evaluated for signs of respiratory distress or exercise intolerance. The functional grading system has since been implemented as a health scheme as the official Respiratory Function Grading Scheme run by the UK Kennel Club (KC) and the University of Cambridge in French Bulldogs, Bulldogs and Pugs, and is currently used in many countries across the world enabling engagement of breeders and pet owners [19].

Rising popularity of the three popular breeds has led to increased concern about the prevalence brachycephalic-related diseases associated within the wider

brachycephalic dog population [20,21]. Alongside BOAS, health issues include ocular disease, skin fold dermatitis, dental malocclusions, dystocia, and neurological pathology. Whilst much of the current research published on BOAS has focussed on the three most popular breeds (French Bulldogs, Bulldogs and Pugs), there are many dog breeds that can be considered to be brachycephalic [16,22]. The terminology has multiple definitions, and different measurement methodologies. Often the term brachycephalic is considered synonymous with 'flat-faced' describing a shortened muzzle, also referred to as facial hypoplasia. The craniofacial ratio measures this (snout length is divided by circumferential cranial length), and ratios of less than 0.3 are considered extremely brachycephalic, with ratios between 0.3 to 0.5 being moderately brachycephalic [23]. Though facial hypoplasia is a manifestation of brachycephaly, the term is derived from the Greek for 'short head' and therefore describes the morphology of the whole skull. Dogs that have a relatively wide skull in comparison to the length, such as the Staffordshire Bull Terrier can be considered brachycephalic under this definition, also referred to as the cephalic index [24]. Brachycephaly can also be presented in shortening of the cranial vault along the longitudinal axis. In this respect, increasing brachycephaly in the Cavalier King Charles Spaniel and other toy breeds increase the risk of cerebrospinal fluid disorders and Chiari-like malformation [25–28]. Other indices to measure brachycephaly include calculation of cephalic index (skull width divided by skull length) and craniofacial angle which can be affected by mandibular prognathism [24,29]. The variations in brachycephalic skull morphologies appear to be associated with different brachycephalic disease; however, this is currently poorly defined.

BOAS has been noted to affect other brachycephalic breeds such as the Boston Terrier, Shih Tzu and Pekingese [16,30,31]. Cavalier King Charles Spaniels have also been reported to suffer BOAS alongside sleep disordered breathing [32,33]. Whilst BOAS is associated with brachycephaly, currently there is limited evidence as to how the disease varies between the wide range of phenotypes of different brachycephalic breeds, and the prevalence in other breeds has not yet been documented. This may be because BOAS is less common within these breeds, or possibly that the presentation of the disease differs, resulting in underdiagnosis. Additionally, the populations of some of these breeds are much smaller therefore BOAS affected dogs are seen more infrequently. Recently, stertor and reduction in nasopharyngeal dimensions was recorded in some breeds not classically considered to be brachycephalic [34]. However, the boundary between brachycephalic and mesocephalic conformation is not clear-cut.

Craniofacial ratio (CFR) has been proposed as an important conformational risk factor for BOAS in a previous study across multiple breeds [23], in that more extremely flat-faced breeds are at greater risk. However, sample sizes were small in the breeds investigated. In another study looking at breed-specific models with larger sample sizes, craniofacial ratio was not a significant factor in the differentiation of BOAS status of Pugs or Bulldogs [9]. In that study, nostril stenosis was found to be an important factor associated with BOAS, alongside other conformational factors such as neck girth ratio (NGR). The relationship between relative muzzle length and BOAS risk both across and within breeds therefore may be more complex than originally thought, and variations in the morphology and genetic profiles of different breeds could result in different propensities to the disease.

The aim of this study is to investigate the prevalence of BOAS in different brachycephalic breeds. Fourteen brachycephalic breeds were selected to capture a broad range of brachycephalic phenotypes, including both extreme and more moderate conformations. Due to practical constraints of the study, the list was not intended to be exhaustive. All dogs were were evaluated for BOAS based on the existing respiratory function grading criteria. Signalment factors and conformation measurements were recorded in order to evaluate the key risk factors for BOAS both across and within each breed. The 14 breeds selected for investigation include the Affenpinscher, Boston Terrier, Boxer, Cavalier King Charles Spaniel (CKCS), Chihuahua, Dogue de Bordeaux (DDB), Griffon Bruxellois, Japanese Chin, King Charles Spaniel (KCS), Maltese, Pekingese, Pomeranian, Shih Tzu and Staffordshire Bull Terrier (SBT).

## Methodology

### Subjects

Fourteen brachycephalic breeds were prospectively recruited from between September 2021 to April 2024 from the pet dog, breeding and showing population (n = 898). Dogs were either assessed during individual appointments for participation at the Queen's Veterinary School Hospital (QVSH) in Cambridge or recruited at dog shows or breed-specific health testing days. Dogs included in the study were over 12 months of age. Dogs reported by the owner to have had previous BOAS surgery were excluded. The study was performed under ethical approvals CR530 and CR562 from the Department of Veterinary Medicine, University of Cambridge. Owners were asked to read the study information and written consent was gained through a consent form.

### Respiratory function grading

Respiratory function grading was performed in all dogs using the methodology established in the three popular breeds (Bulldogs, French Bulldogs and Pugs) that had been previously validated using whole-body barometric plethysmography [17,19]. Thoracic auscultation was performed and if any clinical signs or history of lower airway disease was apparent, the subject was excluded from the study. Pharyngo-laryngeal auscultation was performed before and after a 3-minute exercise test whereby the dog is run at a fast trot to a gallop, dependent on their individual speed capability and temperament. The objective was to auscultate the dogs when at rest and again when in minor respiratory deficit. Audible upper respiratory noises described as stertor (low-pitched sounds) and stridor (high-pitched sounds) were graded as none, mild, moderate, or severe. If nasal stridor was heard, rather than laryngeal stridor, this was recorded in the study notes. Following exercise, the dog was also observed for any signs of breathing difficulty or dyspnoea including abdominal effort or cyanosis.

Dogs that completed the exercise test with no dyspnoea or respiratory noise detected were classified as Grade 0, whilst dogs that displayed any upper airway noises were classified as from Grades 1–3 according to the criteria in Liu et al 2016 [18]. Grade 1 is considered mild BOAS whereby stertor or stridor is only detectable with stethoscope auscultation and there are no signs of being clinically affected. Grade 2 is assigned to dogs whereby respiratory noise is audible without a stethoscope. Grade 3 is the most severe status, and is when BOAS results in an inability to exercise or signs of respiratory distress, such as cyanosis. Grade 2 and Grade 3 BOAS grades are considered to be clinically significant. Nostril stenosis was also graded based on the existing published criteria by Liu et al 2017 [9]. A general clinical examination was also performed at the same time.

### Conformational measurements

Conformational measurements were taken as described in a previous study using soft tape and photographic methodology [35]. These are detailed in Table 1. Soft tape measurements include body length (BL), body height (BH), neck girth (NG), chest girth (CG) and tail length (TL). Body length to body height ratio was calculated to give an indication of proportional length of body (BL:BH). Neck to chest girth ratio (NGR) was calculated by dividing neck girth by chest girth (NGR = NG/CG). The photographic measurements were intercanthal distance (ICD) and skull width (SW), used to calculate eye width ratio (EWR = ICD/SW), and muzzle length (MzL) and cranial length (CrL) was was used to calculate craniofacial ratio (CFR = MzL/CrL) [9,23]. CFR and EWR are displayed in Fig 1.

### Statistical methods

**BOAS grade distribution.** In 2016, a study was conducted on French Bulldogs, Bulldogs and Pugs that reported the BOAS grade distribution within the sample population for each breed [18]. The reported proportion of Grade 0 dogs for each breed was used as a comparison to the percentage of BOAS unaffected dogs in this study. A one-sided Fisher's exact test

**Table 1. Conformational measurements taken through soft tape and photographic measurements.**

| Measurement | Method of collection | Anatomical landmarks |
|---|---|---|
| **Body length (BL)** | Soft tape | Distance from cranial edge of scapula to tail root on the dorsal midline. |
| **Body height (BH)** | Soft tape | Distance from cranial edge of scapula (on midline) down forelimb to level of floor. |
| **Neck girth (NG)** | Soft tape | Mid-circumferential girth of neck. |
| **Chest girth (CG)** | Soft tape | Girth measured at deepest level of chest. |
| **Tail length (TL)** | Soft tape | Distance from root of the tail to the most caudal edge. |
| **Skull width (SW)** | Frontal plane and dorsal plane photographs | Linear distance between the most lateral points of the zygomatic arches. |
| **Intercanthal distance (ICD)** | Frontal plane photographs | Linear distance between the medial canthi. |
| **Muzzle length (MzL)** | Sagittal plane photographs | Linear distance from the stop to the rostral edge of nasal planum. |
| **Cranial length (CrL)** | Sagittal plane photographs | Circumferential distance from the stop to the occiput in midline sagittal plane. |

Table reprinted from Tomlinson F, O'Neill E, Liu NC, Sargan DR, Ladlow JF. BOAS in the Boston Terrier: A healthier screw-tailed breed? PLOS ONE. 2024 Dec 31;19(12):e0315411.

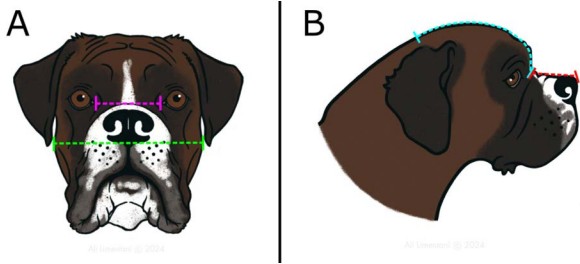

**Fig 1. Photographic conformational measurements. (A)** Eye width ratio calculated from frontal plane photographic measurements (intercanthal distance (pink line) and skull width (green line)). **(B)** Craniofacial ratio calculated from sagittal plane photographic measurements (muzzle length (red line) and cranial length (blue line)). Reprinted under a CC BY license, with permission from Ali Limentani, original copyright 2024.

was used to test the hypothesis that the brachycephalic breeds in this study have a lower prevalence of BOAS than the three most popular breeds. Because multiple pairwise comparisons were made between each breed and French Bulldogs, Bulldogs and Pugs, the p values were adjusted using the Bonferroni-Dunn method with alpha set at 0.05 and the threshold of $p < 0.001$ considered significant. Breeds with less than 25% Grade 0 dogs, comparable to rates reported in the three popular breeds, were considered 'high BOAS risk' breeds. Breeds with 25–50% Grade 0 dogs were labelled 'moderate BOAS risk'. Whilst breeds with more than 50% of dogs Grade 0 and some dogs graded 1–3, were considered 'mild BOAS risk'.

**Factors associated with BOAS.** Collinearity of the conformation measurements (BH, BL, BL:BH, TL, NGR, EWR, CFR) was checked prior to multivariate analysis using Pearson r. Multiple logistic regression was performed with the signalment factors and conformational measurements that did not show collinearity (age, sex, neutering status, BCS, nostril stenosis, BL:BH, TL, NGR, CFR). Categorical variables were transformed. Nostril stenosis was coded as: 0 = open, 1 = mild stenosis, 2 = moderate stenosis, 3 = severe stenosis. Body condition score (BCS) was classified in two groups: 0 = normal or underweight (BCS ≤ 5) and 1 = overweight (BCS ≥ 6). Classification cut-off was set at 0.5. Backwards

stepwise elimination was performed to produce the final model. Breed-specific models were developed using the final model variables: BCS, nostril stenosis, and CFR.

Individual statistical tests were performed on each conformational parameter comparing Grade 0 dogs with Grade 1–3 dogs in each breed. A one-tailed student's t test (unpaired, parametric) was used for continuous variables (BL, BH, BL:BH, NGR, TL, EWR, CFR) and the F test performed to test for equality of variance. Welch's correction was applied to the variables that did not demonstrate equality of variance. In analysis of binary categorical variable data, a two-sided Fisher's exact test was performed. Body condition score was classified under binary data whereby normal or underweight (BCS ≤ 5) was compared to overweight scores (BCS ≥ 6). Binary outcomes were present in nostril stenosis for CKCS, Pomeranians and SBT (open or mild nostril stenosis). For the other breeds, nostril stenosis was considered ordinal data and therefore a Cochrane-Armitage test for trend was performed. In the univariate analysis, Holm-Sidak corrections were applied setting p-value significance at 0.004 to allow for multiple comparisons. Simple logistic regression was performed on the CFR data in two ways: firstly, of the whole sample data, and secondly with each individual breed.

**Statistical software.** Statistical analysis was performed almost exclusively using GraphPad Prism version 9.5.0 for Mac OS, GraphPad Software, San Diego, California USA, www.graphpad.com. R (version 4.5.0 (2025-04-11); R Foundation for Statistical Computing, Vienna, Austria) within RStudio (version 2024.12.1 + 563) was used in the simple logistic regression analysis of CFR with the function glm of the stats package (version 4.5.0), ggplot2 package (version 3.5.2) and pROC package (version 1.18.5). In this paper, results described as significant indicate a p-value of $p < 0.05$ unless otherwise stated.

## Results

### Sample population characteristics

A total of 898 dogs were recruited into the study of the fourteen different breeds (Affenpinscher n = 69, Boston Terrier n = 107, Boxer n = 79, CKCS n = 73, Chihuahua n = 47, Dogue de Bordeaux n = 51, Griffon Bruxellois n = 52, Japanese Chin n = 46, KCS n = 83, Maltese n = 32, Pekingese n = 46, Pomeranian n = 51, Shih Tzu n = 42, SBT n = 120). Mean age was 4.7 years old (standard deviation (SD) 3.0) ranging from 1 to 15 years old. Female and male neutered dogs made up 19% and 10% of the sample population respectively, whilst 35% dogs were intact females and 30% were intact males. In 10 males and 15 females, neutering status was unknown or not recorded. Multiple logistic regression analysis found that sex (p = 0.33) and neuter status (p = 0.05) were not significantly correlated with BOAS status. However, across the entire study population, age was positively associated with BOAS with an odds ratio of 1.08 (95% CI: 1.01–1.16, p = 0.02). Signalment characteristics for each of the fourteen brachycephalic breeds can be found in S1 Table. Additionally, concurrent health issues found at the time of clinical examination are also noted in S2 Table.

### Comparative proportions of BOAS Grade 0 dogs

All dogs underwent respiratory function grading assessment. Within each breed, the proportion of Grade 0 dogs was compared to that of the three popular brachycephalic breeds (Pugs 7%, French Bulldogs 10%, and Bulldogs 10.9%) as shown in Fig 2 [18]. Eight dogs were omitted from the analysis as they did not complete the exercise tolerance test, due to either behavioural reasons or comorbidities such as orthopaedic, neurological, or cardiopulmonary disease. Two breeds were identified as having no significant difference in prevalence of BOAS to the three popular breeds: the Pekingese (adjusted p = 1.00) and Japanese Chin (adjusted p = 1.00), therefore described as higher risk breeds. A number of other breeds were defined as a moderate risk of BOAS with between 25–50% of dogs being Grade 0; the Griffon Bruxellois, Boston Terrier, Dogue de Bordeaux, King Charles Spaniel and Shih Tzu. A further number of breeds had at least 50% dogs being Grade 0 and at least 75% clinically unaffected (Grade 0 and Grade 1); Staffordshire Bull Terrier, Cavalier King Charles Spaniel, Chihuahua, Boxer, Affenpinscher, Pomeranian. Within the Maltese sample population only one dog tested as Grade 1,

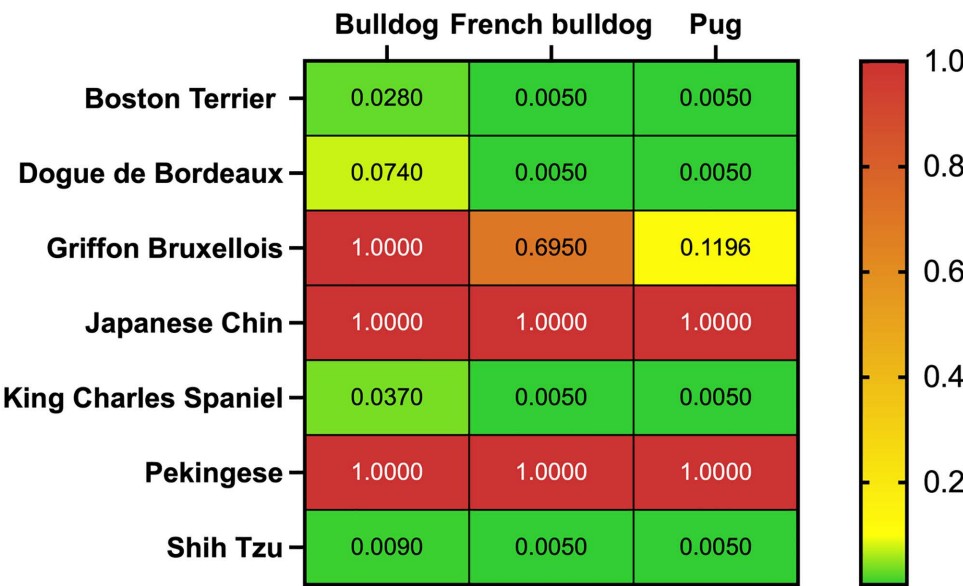

## Pairwise Comparison P values

**Fig 2. P values of pairwise comparisons between the proportion of Grade 0 dogs in brachycephalic breeds that had a p value >0.005 when compared to Pugs, French Bulldogs or Bulldogs.** Adjusted p-values of one-sided Fisher's exact test to test the alternative hypothesis that the fourteen brachycephalic breeds in this study have a greater proportion of dogs scoring Grade 0 in the respiratory function assessment. All other breeds studied (Affenpinscher, Boxer, Cavalier King Charles Spaniel, Chihuahua, Maltese, Pomeranian, Staffordshire Bull Terrier) had significantly greater proportion of Grade 0 dogs in the sample population with p-values of 0.005 against all 3 breeds.

and due to a smaller sample size (n = 32) this breed was excluded from the singular breed conformation risk factor analysis. The percentages of the BOAS grades in each breed are displayed in Fig 3 and detailed in S3 Table. Additionally, whilst few Affenpinschers were found to be BOAS affected, 26% were found to show signs of tracheal collapse; having a characteristic 'honking' cough.

### Conformational factors associated with BOAS

Three conformational variables were noted significant in multivariate analysis of the combined breed population: body condition score, nostril stenosis and craniofacial ratio. The model estimates are displayed in Table 2 and the final model is displayed in Fig 4. Moderate and severe nostril stenosis were shown to have a significant positive association with BOAS status ($p < 0.001$ and $p = 0.03$, respectively). Craniofacial ratio was negatively associated with BOAS status indicating that dogs with a lower CFR were more likely to have BOAS. Body condition score was also positively associated with BOAS, with an odds ratio of 1.8 (95% CI: 1.3–2.6). These three variables incorporated in the final model result in correct classification of 79% BOAS Grade 0 dogs and 60% BOAS Grade 1–3 dogs. Tjur's R squared was 0.2 therefore the model accounts for approximately 20% of the variation between BOAS status in these breeds. Mean conformational variables for all measurements can be found in S1 Fig.

### Breed-specific models

Breed-specific models were plotted using multiple logistic regression. Pomeranians were excluded from this analysis as there were too few Grade 1–3 dogs (n = 8) for the data to be modelled. Quasi-perfect separation was present in certain

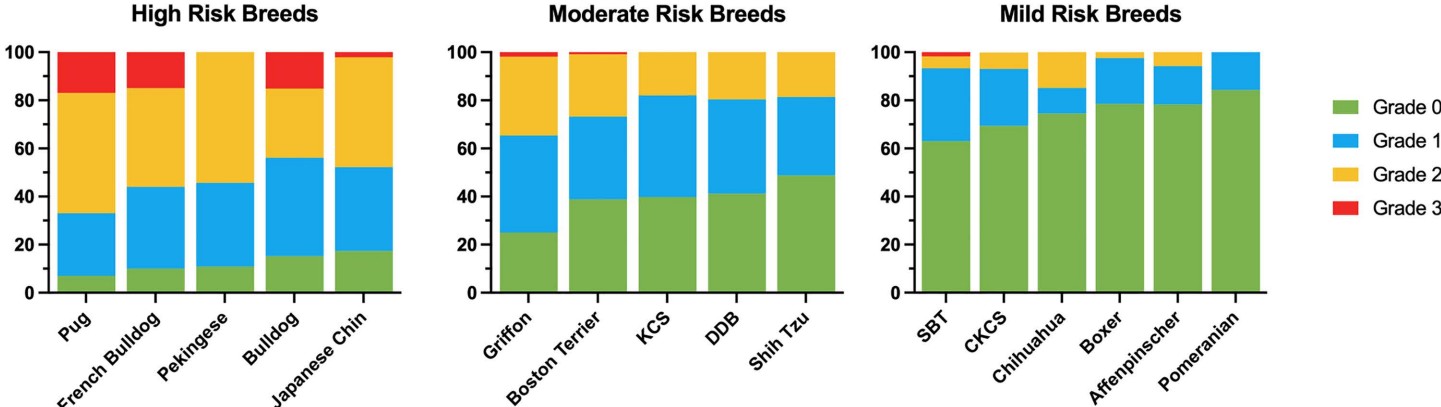

**Fig 3. BOAS grade distribution across the breeds tested in comparison to the 3 most popular breeds.** Pug, French Bulldog and Bulldog grade distributions taken from previous study (18). **(A)** High risk breeds: Pekingese (n = 45) and Japanese Chin (n = 46). **(B)** Moderate risk breeds: Griffon Bruxellois (n = 52), Boston Terrier (n = 107), King Charles Spaniel (n = 83), Dogue de Bordeaux (n = 51) and Shih Tzu (n = 42). **(C)** Mild risk breeds: Staffordshire Bull Terrier (n = 120), Cavalier King Charles Spaniel (n = 73), Chihuahua (n = 47), Boxer (n = 79), Affenpinscher (n = 69), and Pomeranian (n = 51). Maltese data not included in this figure as number of affected dogs insufficient to draw reasonable conclusion (total n = 32, Grade 1 n = 1).

**Table 2. Multiple logistic regression analysis final model. Body condition score [1]: ≥ 6 compared to reference level [0]: ≤ 5. Nostril stenosis [1]: mild stenosis, [2]: moderate stenosis and [3]: severe stenosis compared to reference level [0]: open nostrils.**

| Variable | Parameter estimates | | | Odds ratios | | |
|---|---|---|---|---|---|---|
| | Estimate | SE | 95% CI | Estimate | 95% CI | p value |
| Intercept | 0.36 | 0.25 | −0.13-0.85 | 1.4 | 0.87-2.3 | 0.16 |
| BCS [1] | 0.6 | 0.19 | 0.24-0.97 | 1.8 | 1.3-2.6 | 0.001 ** |
| Nostrils [1] | 0.37 | 0.19 | −0.015-0.75 | 1.4 | 0.99-2.1 | 0.06 |
| Nostrils [2] | 1.4 | 0.3 | 0.86-2.0 | 4.2 | 2.4-7.7 | <0.001 *** |
| Nostrils [3] | 2.3 | 1.1 | 0.58-5.3 | 10 | 1.8-191 | 0.03 * |
| CFR | −5.4 | 0.85 | −7.1-3.7 | 0.0047 | 0.0009-0.025 | <0.001 *** |

variables in the datasets of the Chihuahua (nostril stenosis) and Japanese Chin (nostril stenosis and BCS), therefore these variables were excluded in these two models. Specific variables were found to be statistically significant in six breed models. BCS was significant in three models: Affenpinscher (OR 9.7, 95% CI: 1.4–194, p = 0.046), CKCS (OR 2.2, 95% CI: 1.2–13.7, p = 0.025) and Shih Tzu (OR 5.0, 95% CI: 1.1–28.9, p = 0.047). CFR was significant in three models: Boston Terrier (OR 4.8e-013, 95% CI: 3.1e-021 - 5.3e-006, p = 0.0014), Boxer (2.7e-011, 95% CI: 1.6e-021-0.01, p = 0.026), and Chihuahua (OR 9.8e-010, 95% CI 2.3e-018-0.005, p = 0.020). Nostril stenosis was significant in the Boston Terrier model (OR 2.3, 95% CI 1.2–4.7, p = 0.017). The plotted receiver operator curves for each breed model can be found in S2 Fig.

## Body condition score

Body condition score was noted as significant in the across breed multivariate analysis. At an individual breed level, there are differences in the spread of underweight, normal, and overweight dogs as shown in a small number of examples in Fig 5. In the multivariate breed-specific models, BCS was statistically significant in the Affenpinscher, CKCS and Shih Tzu. In the univariate analysis, revealed BCS was significant in raw p values of the CKCS (p = 0.006), the KCS (p = 0.02), and

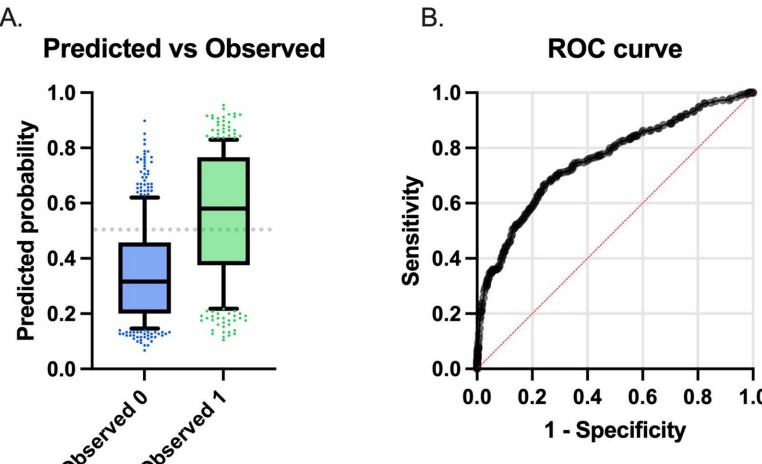

**Fig 4. Final model from the multiple logistic regression analysis incorporating BCS, nostril stenosis and CFR. (A)** Predicted probability of the model correctly classifying Grade 0 (Observed 0) or Grade 1−3 dogs (Observed 1). Negative predictive power of 71% and positive predictive power of 69%. **(B)** Area under ROC curve = 0.75 (95% CI: 0.72-0.79).

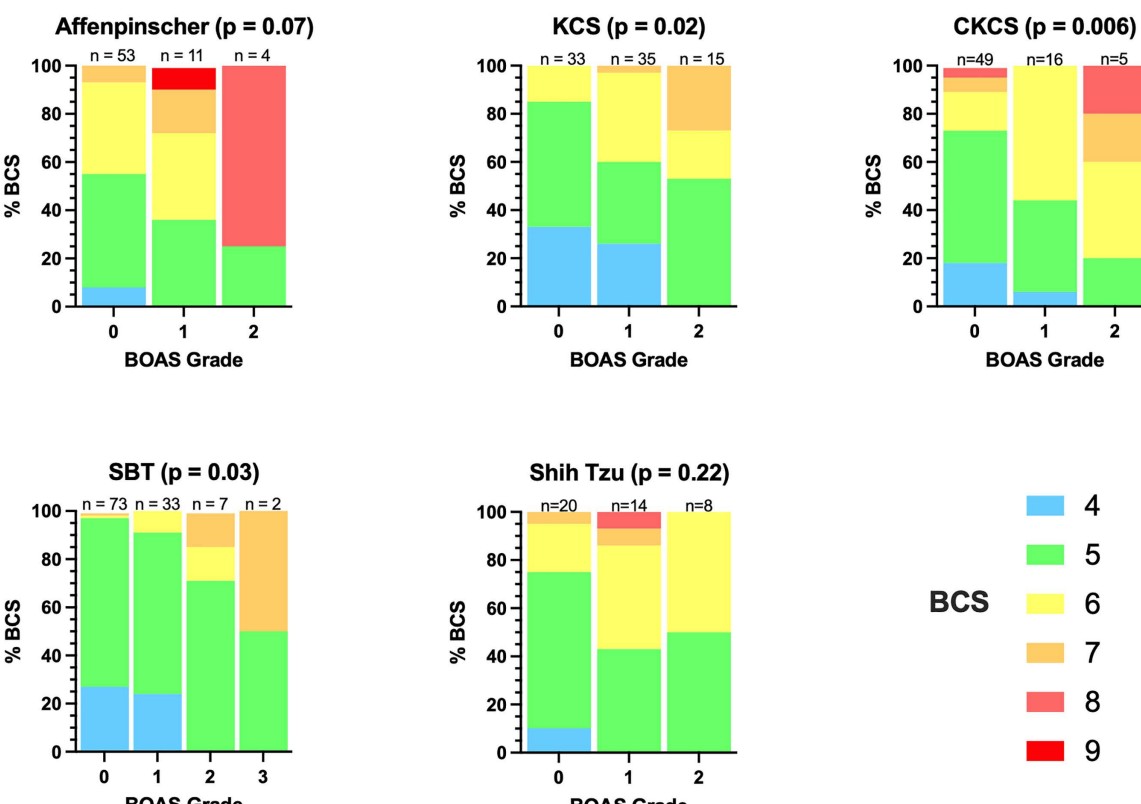

**Fig 5. Percentage distribution of body condition score by BOAS Grade five breeds with statistical significance.** P-values indicate the raw p-value from the two-sided Fishers-exact test comparing normal or underweight (BCS ≤ 5) to overweight (BCS ≥ 6). Included are breeds with a statistical significance from the multivariate breed-specific models (Affenpinscher and Shih Tzu).

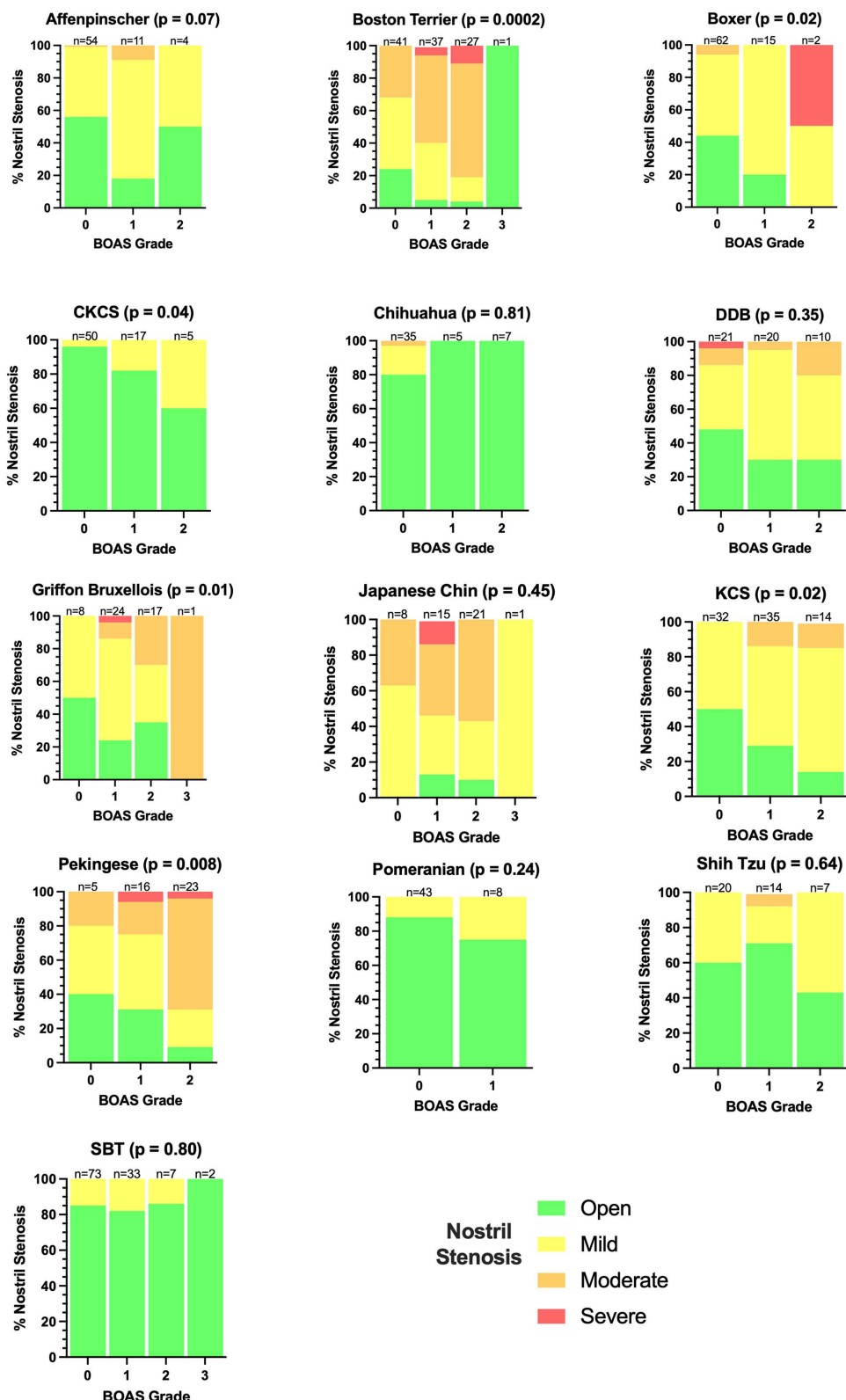

**Fig 6. Percentage distribution of nostril stenosis severity across different brachycephalic breeds by BOAS Grade.** P-values indicate the raw p-value obtained in univariate analysis by two-sided Fisher's exact test for binary outcomes (CKCS, Pomeranian and SBT) or Cochrane-Armitage test for trend (remaining groups).

the Staffordshire Bull Terrier (p = 0.03), However these were not significant when the p-values were adjusted using the Holm-Sidak method (see supplementary information S4 Table). Additionally, though not included in statistical analysis, it was noted that the single Maltese dog that was classified a Grade 1 dog had a BCS of 7/9.

### Nostril stenosis

Nostril stenosis was found to be significantly correlated with BOAS in the multivariate analysis performed across all fourteen breeds. The distribution of nostril stenosis severity is displayed in Fig 6. The two breeds found to be high risk for BOAS, the Pekingese and Japanese Chin, have high rates of nostril stenosis. Two breeds at moderate risk, the Griffon Bruxellois and the Boston Terrier also have higher rates of nostril stenosis compared to the other breeds. Severe nostril stenosis has only been recorded in six breeds (Boston Terrier, Boxer, Dogue de Bordeaux, Griffon Bruxellois, Japanese

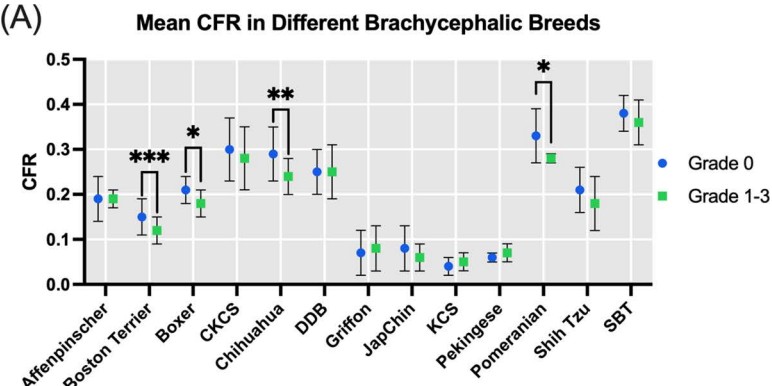

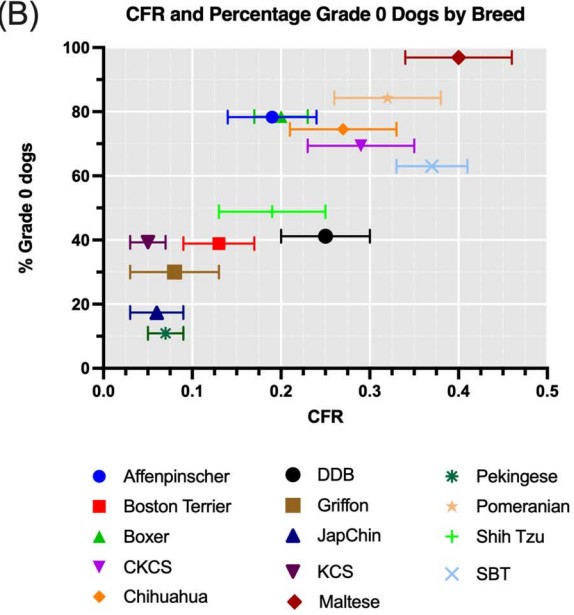

**Fig 7. Craniofacial ratio values across the fourteen brachycephalic breeds included in the study. (A)** Mean craniofacial ratio in each breed comparing Grade 0 to Grade 1-3 dogs. Error bars indicating standard deviation. Significant results annotated *. **(B)** Mean craniofacial ratio for each breed plotted against the percentage of Grade 0 dogs within that breed. Error bars indicate standard deviation.

Chin, Pekingese). Certain breeds show none or very few dogs with moderate (CKCS, Pomeranian, Staffordshire Bull Terrier) or severe stenosis (Affenpinscher, Chihuahua, Maltese).

## Craniofacial ratio

Craniofacial ratio was found to be a significant conformational factor negatively associated with BOAS status in the multivariate analysis across the fourteen breeds. The mean craniofacial ratio for each breed has been plotted for the Grade 0 and Grade 1–3 groups in Fig 7. Within the intra-breed analysis, in comparisons between Grade 0 and Grade 1–3 dogs, the difference in craniofacial ratio was significant in unadjusted p-values of the Boston Terrier (0.15 vs. 0.12, p < 0.0001), Boxer (0.21 vs. 0.18, p = 0.02), Chihuahua (0.29 vs. 0.24, p = 0.004), and Pomeranian (0.33 vs. 0.28, p = 0.0001). For the Maltese, there was a significant difference between Grade 0 dogs and the single Grade 1 dog (p = 0.01) with an average of 0.4 CFR compared to 0.26 in this dog. Within the Shih Tzu (0.21 vs. 0.18, p = 0.07) and Staffordshire Bull Terrier (0.38 vs 0.36, p = 0.06), the p value was approaching significance. There was no significant difference between the CFR of Grade 0 and Grade 1–3 groups of breeds that had on average a CFR of < 0.1 (KCS p = 0.35, Japanese Chin p = 0.24, Pekingese p = 0.29, Griffon Bruxellois p = 0.18), however between these breeds the proportion of Grade 0 dogs varies from 10.9% in the Pekingese to 39.8% in the KCS. Full raw and adjusted p-values can be found in Supplementary Information S4 Table.

A simple logistic regression model was plotted to compare the effect of CFR on BOAS status across the whole study sample, as shown in Fig 8. Model A describes the probability of BOAS Grade 0 compared to BOAS Grades 1–3 at different CFR values. The value of Tjur's R squared is 0.16 for the model. This indicates how craniofacial ratio only accounts for approximately 16% variance in having audible upper respiratory noise (Grade 1 and above). Individual breed logistic regression analysis results and plotted curves can be found in S3 Fig and S5 Table.

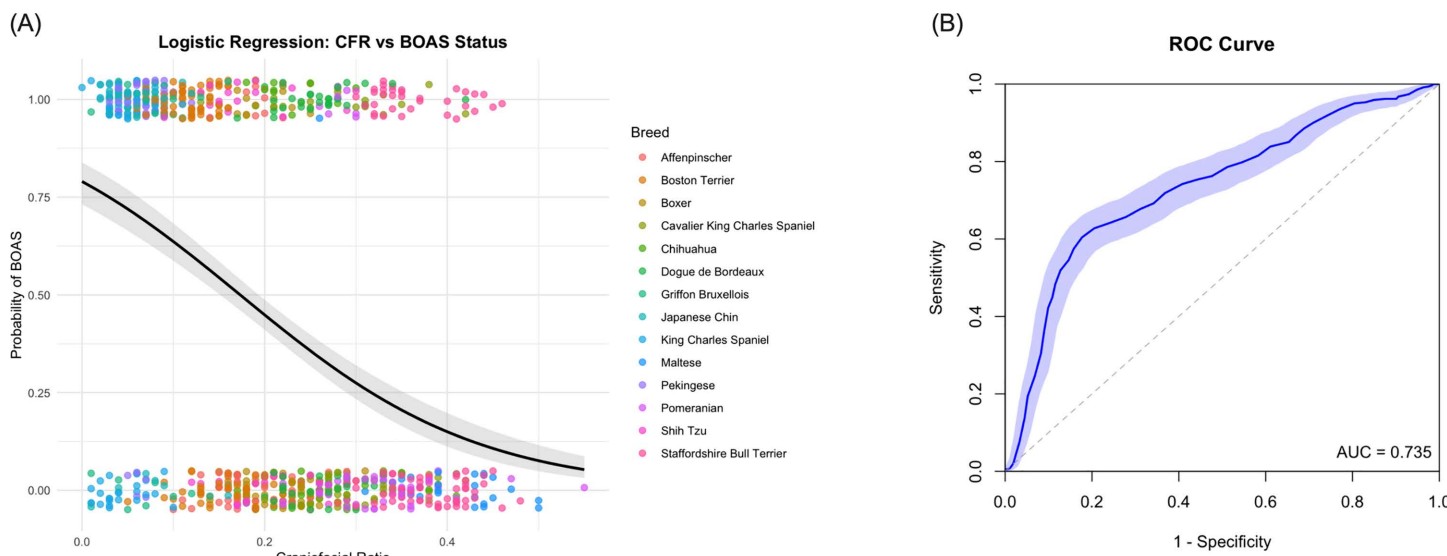

**Fig 8. Simple logistic regression model plotted from the fourteen brachycephalic breeds indicating the probability of BOAS grades at different craniofacial ratios. (A)** Craniofacial ratio plotted against the probability of being BOAS Grades 1−3 versus Grade 0. Grey area indicates the 95% confidence interval. **(B)** Receiver Operator Characteristic (ROC) curve. Area under ROC curve = 0.74 (95% CI: 0.70-0.77, p < 0.0001).

## Univariate intra-breed analysis: Other conformational factors

The univariate analysis results for the within breed analysis can be found in S4 Table. Within the univariate analysis, few results were noted as significant following adjustment of p-values. Tail length was noted as significantly associated with BOAS for the Shih Tzu (adjusted p = 0.04) and Staffordshire Bull Terrier (adjusted p = 0.004). Shorter tails were associated with an increased risk of BOAS in these breeds. Prior to adjustment, neck girth ratio was noted as significant in the Boston Terrier (p = 0.04) and Staffordshire Bull Terrier (p = 0.007) with dogs with proportionately thicker necks more likely to be affected, and BL:BH ratio was significant in the Chihuahua (p = 0.047) and King Charles Spaniel (p = 0.02) in that dogs with relatively longer bodies with a shorter height were more likely to be BOAS affected. In two breeds, a shorter height was positively associated with BOAS status (CKCS p = 0.02, Shih Tzu p = 0.01). A larger EWR was associated with BOAS status in the Boxer (p = 0.03) whilst a smaller EWR was in the Shih Tzu (p = 0.04).

## Discussion

This cross-sectional study demonstrates that the prevalence of BOAS varies considerably between the different brachycephalic breeds. Two breeds are identified as being at a high risk of BOAS: the Pekingese and Japanese Chin. These breeds have comparable rates of disease to the three popular brachycephalic breeds (the Pug, French Bulldog and Bulldog) in which BOAS has been reported on considerably. There are five breeds in which a moderate risk of BOAS is identified: the Griffon Bruxellois, Boston Terrier, King Charles Spaniel, Dogue de Bordeaux and Shih Tzu. These five breeds have significantly more Grade 0 dogs within the breed than the three popular breeds, but at least 50% of dogs in the sample population are Grades 1–3.

The remaining breeds have a substantial portion of the sample population (over 50%) assessed as Grade 0, with few dogs being clinically affected by BOAS. This includes the SBT, CKCS, Chihuahua, Boxer, and Affenpinscher. These dogs are described as having a mild risk of BOAS. There were no clinically affected (Grade 2 or 3) dogs within the Pomeranian or the Maltese sample population. Within the Maltese, only one dog was identified as a Grade 1. However, this was in a smaller sample population compared to the other breeds (n = 32), therefore there is limited evidence to draw conclusions regarding the risk of BOAS within this breed. In a small Korean study, three of 17 Maltese dogs enrolled from a pet population visiting a veterinary centre but not admitted for respiratory disease had stertorous breathing [34], therefore prevalence may vary between different populations.

Three key conformational risk factors were identified following multivariate analysis across the whole dataset; BCS, nostril stenosis and CFR. Further intra-breed analysis revealed the variations within breeds, whereby only four breeds had significant differences of the CFR between Grade 0 and Grade 1–3 groups: the Boston Terrier, Boxer, Chihuahua, and Pomeranian. The most extremely flat-faced breeds with an average CFR of < 0.1 (KCS, Japanese Chin, Pekingese, Griffon Bruxellois) did not show a significant difference between Grade 0 and Grade 1–3 dogs. This may be due to a combination of intra-breed variation, effect of measurement error or small effect size. An unexpected finding was noted in that for one extremely flat-faced breed, the KCS (mean CFR of 0.05), 40% of dogs were Grade 0. Certain dog breeds were found to have body condition score as a significant factor for BOAS grade. This includes the Affenpinscher, CKCS, and Shih Tzu. In these three breeds, weight loss could be used as a management tool to reduce the risk of BOAS. In other breeds, few overweight dogs were noted. For example, there were few overweight Pekingese despite being in the 'high risk' category. In this case, it is likely that other factors are more impactful on BOAS status.

Nostril stenosis has previously been reported as a key risk factor for BOAS [9,23]. Within this study, the prevalence of nostril stenosis has been found to vary substantially in different brachycephalic breeds. Across the breeds, it is significantly associated with increased BOAS risk, whilst in the breed-specific analysis, a significant effect was only found in Boston Terriers. This may be due to sample sizes limitations, or due to less variation of nostril stenosis within breeds. Within the high-risk breeds, the Japanese Chin and Pekingese have some dogs with severe stenosis recorded, and an overall higher prevalence of nostril stenosis (only 5.8% and 17.8% of dogs respectively had open nostrils). Other moderate risk breeds such as the Griffon Bruxellois, Boston Terrier and Dogue de Bordeaux also have severe stenosis recorded.

One Boxer was recorded with severe nostril stenosis and had been assessed as a Grade 2, indicating that whilst rare in this breed, severe stenosis can occur and is likely to be associated with respiratory signs.

Other risk factors have been identified within the intra-breed analysis. Neck girth ratio has previously been noted to be a significant factor in the risk of BOAS in Bulldogs and French Bulldogs [9]. In this study, there was some evidence that this is a conformational risk factor for Boston Terriers and SBTs in that dogs with relatively thicker necks are more likely to be affected by BOAS. Increased neck circumference is associated with sleep apnoea in humans [36]. The similarities regarding this between SBTs, Boston Terriers and the bulldog breeds are of note when considering the close genetic relationship between these breeds [37]. These are features of 'screw-tailed' dog breeds, and the variant allele *DVL2* gene causing this phenotype is fixed or close to fixed in the French Bulldog, Bulldog and Boston Terrier, and can occur in the SBT [38]. Similarly, there was a significant association with BOAS for dogs with shortened tail length in the Staffordshire Bull Terrier and Shih Tzu, and close to significance in the Boston Terrier.

Conformational risk factors can be useful for both breeders and prospective owners in selecting dogs which are less likely to be affected by BOAS. Knowledge of these risk factors can also help to inform breed standards in deciding which features are detrimental to health so that factors associated with BOAS are not rewarded in the show ring, particularly as winning dogs can become popular sires [39]. However, the risk factors presented in this analysis indicated that the model of the three conformation factors (BCS, nostril stenosis and CFR) together only accounted for 20% of variation in BOAS status across the different breeds. Additionally, CFR was only responsible for 16% of variation in the difference between Grade 0 and Grade 1–3 affected dogs. There is substantial genetic and phenotypic variation across the brachycephalic breeds, therefore each individual breed has its own risk profile for BOAS and different factors affecting this. With the current understanding, use of the respiratory function grading is more likely to be accurate in determining BOAS status and therefore which dogs should be selected for breeding, or whose welfare would benefit from veterinary intervention. The scope of this study is focussed on BOAS and does not encompass other health conditions that affect brachycephalic dogs. Certain breeds could be more acutely affected by BOAS due to undiagnosed or unrecognised existing comorbidities, therefore evaluating the overall health picture of the dog would be optimal to assess welfare status.

The analysis of the proportion of BOAS grades across the different breeds enables identification of breeds that could benefit from the implementation of interventions to mitigate the impact on welfare. The official Respiratory Function Grading Scheme is currently designed for Pugs, French Bulldogs and Bulldogs. Engagement of the scheme by owners and breeders has raised awareness of the seriousness of the disease, and increased knowledge regarding disease recognition and navigating treatment. Opening avenues for assessing more breeds could enable more dogs to be tested and open up engagement for other brachycephalic health issues in these breeds. The high and moderate risk brachycephalic breeds are likely to have the most to benefit from the introduction of a scheme. The moderate risk breeds have a higher proportion of Grade 0 dogs, and therefore are potentially better placed to breed away from individuals with BOAS.

Within the three most popular breeds, there are a number of variations in the clinical signs and course of disease of BOAS. For example, Pugs are prone to laryngeal collapse and are more likely to present with stridor [6]. However, for the Bulldog, suffering from a hypoplastic trachea is much more common [40]. For the breeds found to be affected by BOAS in this study, further investigation into the pathophysiology and anatomical lesion sites would provide insight into how and why they are affected by BOAS. Genetic studies are also important in identifying variants which can result in airway disease. For example, the Norwich terrier, a mesocephalic breed, suffers from upper respiratory obstruction that cannot be attributed to brachycephalic conformation [41,42]; a genetic study found correlation to a genetic mutation in *ADAMTS3* which was also noted to be prevalent in Bulldogs and French Bulldogs [43]. Further research into the underlying function and structure of airway tissues through histological and cytological techniques may be useful in determining how the tissues function and deteriorate over time. This may help to explain why there is significant variation in the number of affected dogs in the extremely flat-faced breeds (mean CFR < 0.1), for example the KCS (39.8%, Grade 0) compared to

the Pekingese (10.9%, Grade 0). The assumption that the degree of facial hypoplasia is the primary cause of BOAS is likely an oversimplification, and the underlying pathophysiology of BOAS is much more complex.

## Limitations

The models for conformational risk factors did not include data for the three popular brachycephalic breeds (the Pug, French Bulldog and Bulldog). However extensive previous research has been carried out as to how the conformational measurements affected BOAS risk in these breeds [9]. The data for the BOAS grades in these three breeds was taken from a study published in 2016, therefore may not reflect the current BOAS grade distribution of the present day population. It is noted that there is a different distribution of grades for each of these three breeds in the published Kennel Club RFG data [44]. Dogs recruited for this study were volunteered by owners that were willing to attend an appointment in Cambridge, UK or participate at shows or health testing days arranged by breeders within the showing community. The voluntary nature of this study as well as the location and the nature of the events at which samples were taken may have introduced sampling biases, so that the dogs sampled may not be a true representation of the pet population. Certain factors were limited by the sample size of the study. For the intra-breed analysis, both raw and adjusted p-values have been reported. For significance of the adjusted p-values, significance level was set at 0.004. However, due to the sample size of dogs included in this study, only high effect sizes (approximately >0.6) would be significant. Therefore the raw p-values should be considered in this context.

The respiratory function grading assessment used in this study has previously been validated in French Bulldogs, Pugs and Bulldogs through whole-body barometric plethysmography. This study undertakes the assumption that the increased upper respiratory noise is correlated with increased severity of BOAS in other breeds. During undertaking of the study, variation was noted in the presentation of respiratory noise between the different breeds. In the author's experience, affected larger, heavy-set dogs such as SBTs and DDB often make louder, pharyngeal stertorous sounds when open-mouth breathing, compared to smaller breeds such as the Chihuahua, Griffon Bruxellois or Japanese Chin in which quieter, nasal stertors are more common. The RFG has been used in this study as a screening tool for BOAS. Further investigation into the pathological lesion sites of the obstructive noises was beyond the scope of this current study but further work is being undertaken. For the treatment of individual cases of BOAS, an individual in-depth clinical assessment and use of advanced imaging will often be necessary to determine whether surgical intervention is advisable. The sites of airway obstruction can vary within the three more popular breeds, therefore there it is likely that obstructive sites in other breeds will also be breed-specific. The use of a respiratory function grading scheme for more breeds must take into consideration the particularities of different breeds in order to be effective.

## Conclusion

Brachycephalic dog breeds vary substantially by their phenotype. There is currently no single established definition of a brachycephalic dog, and it remains sensible to evaluate the extent to which brachycephalic breeds are affected by health issues by their individual breed. In this study, the prevalence of BOAS varies substantially between brachycephalic breeds. A single conformational factor which determines BOAS status has not yet been found, however there is a significant association with BCS, nostril stenosis and CFR. Even within the most extremely flat-faced breeds, there is variation in how they are affected by BOAS. Further studies into the internal anatomical lesion sites, genetics and histopathology of airway tissues could benefit the understanding of the underlying disease mechanisms and help to explain the variation between individual dogs affected by BOAS.

## Supporting information

**S1 Dataset. BOAS grading and confirmation parameters dataset.**
(XLSX)

**S1 Table. Signalment characteristics for each of the fourteen brachycephalic breeds that underwent respiratory function assessment.**
(DOCX)

**S2 Table. Additional health conditions detected on clinical examination in the fourteen brachycephalic breeds.**
(DOCX)

**S3 Table. BOAS grade distribution recorded for each of the 14 brachycephalic breeds.**
(DOCX)

**S4 Table. P values from within breed univariate analysis comparing BOAS Grade 0 to BOAS Grade 1–3 dogs.** (A) Raw p-values from continuous data univariate analysis from one-tailed t-test and categorical variable Fisher's exact test p-value results. Significance level is set at 0.05. (B) P-values following adjustment using Holm-Sidak method whereby alpha = 0.05 and p-value significance threshold is set at < 0.004.
(DOCX)

**S5 Table. Odds ratios obtained from simple logistic regression analyses performed on each individual breed comparing BOAS status (Grade 0 versus Grade 1–3) and craniofacial ratio.**
(DOCX)

**S1 Fig. Mean conformational measurements for each of the fourteen brachycephalic breeds.** Physical measurements: body height (BH), body length (BL), tail length (TL). Ratios calculated: neck to chest girth ratio (NGR), eye width ratio (EWR) and craniofacial ratio (CFR).
(TIF)

**S2 Fig. Breed-specific multiple logistic regression models plotted using the final model variables body condition score, nostril stenosis and craniofacial ratio.** Receiver operator curves (ROC) labelled with the area under ROC curve (AUC) and 95% confidence interval (95% CI). The Chihuahua model excludes the variable nostril stenosis and the Japanese Chin model excludes the variables body condition score and nostril stenosis.
(TIF)

**S3 Fig. Breed-specific simple logistic regression curves for craniofacial ratio as a predictor of BOAS status.** Breeds found to have a significant association between BOAS status and craniofacial ratio include the Boston Terrier (p < 0.0001), Boxer (p = 0.02) and Chihuahua (p = 0.02).
(TIF)

## Acknowledgments

We thank all the owners and breeders who kindly brought their dogs to participate in this research, and the breed clubs involved in the organisation of testing events. We are also grateful to the staff at the QVSH, Cambridge for support in running the assessments and participant recruitment. Finally, thank you to Ali Limentani for providing the illustrations featured in this manuscript.

## Author contributions

**Conceptualization:** Francesca Tomlinson, Nai-Chieh Liu, David R. Sargan, Jane F. Ladlow.

**Data curation:** Francesca Tomlinson.

**Formal analysis:** Francesca Tomlinson.

**Funding acquisition:** Nai-Chieh Liu, Jane F. Ladlow.

**Investigation:** Francesca Tomlinson, David R. Sargan, Jane F. Ladlow.

**Methodology:** Francesca Tomlinson.

**Supervision:** Nai-Chieh Liu, David R. Sargan, Jane F. Ladlow.

**Validation:** Francesca Tomlinson, David R. Sargan, Jane F. Ladlow.

**Visualization:** Francesca Tomlinson.

**Writing – original draft:** Francesca Tomlinson.

**Writing – review & editing:** Francesca Tomlinson, Nai-Chieh Liu, David R. Sargan, Jane F. Ladlow.

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
