## [Decision Letter · Decision Letter 0]

5 Nov 2025

Dear Dr. Tomlinson,

Thank you for submitting your manuscript to PLOS ONE. After careful consideration, we feel that it has merit but does not fully meet PLOS ONE’s publication criteria as it currently stands. Therefore, we invite you to submit a revised version of the manuscript that addresses the points raised during the review process.

We look forward to receiving your revised manuscript.

Kind regards,

James J Cray Jr., Ph.D.

Academic Editor

PLOS ONE

Journal Requirements:

2. We note that Figure 1 in your submission contain copyrighted images. All PLOS content is published under the Creative Commons Attribution License (CC BY 4.0), which means that the manuscript, images, and Supporting Information files will be freely available online, and any third party is permitted to access, download, copy, distribute, and use these materials in any way, even commercially, with proper attribution. For more information, see our copyright guidelines: http://journals.plos.org/plosone/s/licenses-and-copyright.

3. Please ensure that you refer to Figures 2, 3, 4, 5, 6, and 8 in your text as, if accepted, production will need this reference to link the reader to the figures.

4. We note you have included a table to which you do not refer in the text of your manuscript. Please ensure that you refer to Table 2 in your text; if accepted, production will need this reference to link the reader to the Table.

Reviewers' comments:

Reviewer's Responses to Questions

**Comments to the Author**

1. Is the manuscript technically sound, and do the data support the conclusions?

Reviewer #1: Yes

2. Has the statistical analysis been performed appropriately and rigorously?

Reviewer #1: Yes

3. Have the authors made all data underlying the findings in their manuscript fully available?

Reviewer #1: Yes

4. Is the manuscript presented in an intelligible fashion and written in standard English?

Reviewer #1: Yes

Reviewer #1: The paper is well written, logical and adds to the growing knowledge about brachycephaly. I have minor comments only.

128. Please change in this study to in that study to avoid confusion that you are talking about the referenced one.

139 can you comment briefly why you chose these 14 breeds of all the other slightly brachycephalic ones?

556. Chihuahua is misspelled.

P values are discussed with both capital and lower case. I don’t think it matters, but please be consistent.

**Do you want your identity to be public for this peer review?** For information about this choice, including consent withdrawal, please see our Privacy Policy

Reviewer #1: No

---

## [Author Response · Author response to Decision Letter 1]

5 Dec 2025

Francesca Tomlinson

Department of Veterinary Medicine

University of Cambridge

Madingley Road, Cambridge

CB3 0ES

ft270@cam.ac.uk

07/05/2025

Editor-in-Chief

PLOS One

Dear Editor,

Thank you very much for the opportunity to revise our manuscript, “A cross-sectional study into the prevalence and conformational risk factors of BOAS across fourteen brachycephalic dog breeds” [PONE-D-25-24732]. I appreciate the comments provided by you and the reviewers, which have helped us improve the clarity and quality of the manuscript.

I have carefully considered each comment and revised the manuscript accordingly. Below, we provide a detailed point-by-point response. All changes in the revised manuscript are tracked as requested.

I have also made minor formatting and grammatical edits throughout the manuscript and added references to the Tables and Figures throughout. Additionally, during revision, the manuscript was carefully reviewed, and I identified a small number of remaining values that had not been updated after earlier corrections which had been implemented prior to the initial submission (resolving previous duplicate records or misclassified breeds in the raw data). We have now updated these values in the Results (lines 268-270) and supplementary information Table 3 and 4 (previously 2 and 3). These minor adjustments do not affect the study’s overall findings or conclusions.

In the period since the original submission, there was an extended delay while suitable reviewers were sought. During this additional time, we were able to complete some further descriptive analysis of other clinical findings observed during the examinations. We have included these data as a new Supplementary Table (Table S2). This table presents only descriptive percentages of additional health issues noted in the study population and is intended solely to provide additional context. These data do not alter the study design, primary analyses, results, or conclusions in any way. We have noted this addition in the revised manuscript and hope that including these supplementary details will enhance transparency and provide useful additional information for readers. Please note, the naming of Supplementary Information Tables S2-S4 have shifted to S3-S5, due to this addition.

Point-by-Point Responses

Reviewer #1

1. Comment: 128. Please change in this study to in that study to avoid confusion that you are talking about the referenced one.

Response: Amended.

2. Comment: 139 can you comment briefly why you chose these 14 breeds of all the other slightly brachycephalic ones?

Response: We have now included a comment on the selection of these breeds on lines 137-139. “Fourteen brachycephalic breeds were selected to capture a broad range of brachycephalic phenotypes, including both extreme and more moderate conformations. Due to practical constraints of the study, the list was not intended to be exhaustive.”

3. Comment: 556. Chihuahua is misspelled.

Response: Amended.

4. Comment: P values are discussed with both capital and lower case. I don’t think it matters, but please be consistent.

Response: Amended throughout manuscript.

Editor’s Comments

1. Comment: Please ensure that your manuscript meets PLOS ONE's style requirements, including those for file naming.

Response: Checked requirements met and minor formatting issues amended.

2. Comment: We note that Figure 1 in your submission contain copyrighted images. We require you to either (a) present written permission from the copyright holder to publish these figures specifically under the CC BY 4.0 license, or (b) remove the figures from your submission:

Response: Copyright permission form attached in submission. Amended figure title to include copyright permission statement.

3. Comment: Please ensure that you refer to Figures 2, 3, 4, 5, 6, and 8 in your text as, if accepted, production will need this reference to link the reader to the figures.

Response: Amended. All figures referenced in text.

4. Comment: We note you have included a table to which you do not refer in the text of your manuscript. Please ensure that you refer to Table 2 in your text; if accepted, production will need this reference to link the reader to the Table.

Response: Amended. All tables referenced in text.

5. Comment: If the reviewer comments include a recommendation to cite specific previously published works, please review and evaluate these publications to determine whether they are relevant and should be cited. There is no requirement to cite these works unless the editor has indicated otherwise.

Response: N/A

6. Comment: Please review your reference list to ensure that it is complete and correct.

Response: Reference list hads been checked and minor formatting issues amended.

We hope that the revisions meet expectations required and look forward to hearing from you again. We would like to thank you again for your time and consideration.

Yours faithfully,

Dr Francesca Tomlinson

BOAS Research Group

Dept. of Veterinary Medicine

University of Cambridge

ft270@cam.ac.uk

---

## [Editor Report · Decision Letter 1]

23 Dec 2025

A cross-sectional study into the prevalence and conformational risk factors of BOAS across fourteen brachycephalic dog breeds

PONE-D-25-24732R1

Dear Dr. Tomlinson,

We’re pleased to inform you that your manuscript has been judged scientifically suitable for publication and will be formally accepted for publication once it meets all outstanding technical requirements.

Kind regards,

James J Cray Jr., Ph.D.

Academic Editor

PLOS One
---

## [Editor Report · Acceptance letter]

PONE-D-25-24732R1

PLOS One

Dear Dr. Tomlinson,

I'm pleased to inform you that your manuscript has been deemed suitable for publication in PLOS One. Congratulations! Your manuscript is now being handed over to our production team.

Kind regards,

on behalf of

Dr. James J Cray Jr.

Academic Editor

PLOS One